## METHOD

# MoCHI: neural networks to fit interpretable models and quantify energies, energetic couplings, epistasis, and allostery from deep mutational scanning data

Andre J. Faure[1,5]* and Ben Lehner[1,2,3,4]*

*Correspondence:
andre.faure@crg.eu; bl11@sanger.
ac.uk

[1] Centre for Genomic Regulation
(CRG), The Barcelona Institute
of Science and Technology,
Barcelona, Spain
[2] University Pompeu Fabra (UPF),
Barcelona, Spain
[3] Institució Catalana de Recerca
I Estudis Avançats (ICREA),
Barcelona, Spain
[4] Wellcome Sanger Institute,
Wellcome Genome Campus,
Hinxton, UK
[5] Current Address: ALLOX, PRBB
Building, C/Dr. Aiguader, 88,
08003 Barcelona, Spain

## Abstract

We present MoCHI, a tool to fit interpretable models using deep mutational scanning data. MoCHI infers free energy changes, as well as interaction terms (energetic couplings) for specified biophysical models, including from multimodal phenotypic data. When a user-specified model is unavailable, global nonlinearities (epistasis) can be estimated from the data. MoCHI also leverages ensemble, background-averaged epistasis to learn sparse models that can incorporate higher-order epistatic terms. MoCHI is freely available as a Python package (https://github.com/lehner-lab/MoCHI) relying on the PyTorch machine learning framework and allows biophysical measurements at scale, including the construction of allosteric maps of proteins.

**Keywords:** Deep mutational scanning, Neural networks, Thermodynamic models, Epistasis, Allostery

## Background

A fundamental goal in biology is to understand how natural and synthetic polymer sequences (DNA, RNA, protein) encode their biophysical properties. Achieving this goal will have profound impacts on genetic prediction and biological engineering and allow a deeper understanding of molecular evolution. Recent improvements in the cost and efficiency of DNA synthesis and sequencing have made it possible to quantify the phenotypic effects of large variant libraries in a single experiment (Fig. 1a). The utility of these high-throughput phenotyping methods—variously referred to as deep mutational scanning (DMS), massively parallel reporter assays (MPRAs), or multiplex assays of variant effect (MAVEs)—is evidenced by their wide application for cataloging the effects of natural human genetic variants in the context of disease. However, as a general strategy, the approach of exhaustive phenotyping has clear limitations. First, the universe of all possible biological sequences is enormous, meaning

**Fig. 1** The MoCHI framework and software package for fitting mechanistic models to deep mutational scanning (DMS) data. **a** The fundamental elements of a DMS experiment. First, an input library of sequence variants for a given gene is constructed by direct synthesis or mutagenesis. A competition assay then either physically separates or enriches the output library for sequences with a molecular function of interest. Finally, a quantitative phenotype score is obtained from variant counts before and after selection as determined by high-throughput sequencing. **b** A general framework for fitting custom mechanistic models to DMS data using neural networks. Left: Variant sequences are transformed to energies via the additive trait map $f$; the global epistasis function $g$ describes the nonlinear relationship between the energetic effects of mutations and the molecular phenotype of interest $p$; the experimental measurement process $h$ transforms the molecular phenotype to the specific units of the DMS assay. Right: Graph representation of an example custom model for the inference of two biophysical traits ($\phi_1$ and $\phi_2$) using data from three DMS experiments ($y_1$, $y_2$, and $y_3$) that report on two related molecular phenotypes ($p_1$ and $p_2$). **c** Architecture of the MoCHI software package indicating the four modules handling data management ("data.py"), model definition and fitting ("models.py"), reporting of results ("report.py"), and the implementation of predefined inference workflows ("project.py")

that this approach quickly becomes infeasible even for relatively short polymers. This highlights the need to learn predictive models of variant effects, which can generalize beyond the specific subset of sequences that comprise typical variant libraries. Second, measured phenotype scores do not report directly on the underlying biophysical

**Table 1** Comparison of MoCHI capabilities to previously developed methods

| Feature | LANTERN [20] | MAVE-NN [7] | MoCHI |
|---|---|---|---|
| Global (non-specific) epistasis inference | Yes | Yes | Yes |
| User-specified mechanistic model fitting | No | Yes | Yes |
| Pairwise genetic interaction inference (specific epistasis) | No | Yes | Yes |
| Multidimensional global (non-specific) epistasis inference | Yes | No | Yes |
| Model fitting to multiple (multimodal) phenotypes | Yes | No | Yes |
| Simultaneous fitting of global (non-specific) and pairwise (specific) epistasis | No | Yes | Yes |
| Higher-order genetic interaction inference | No | No | Yes |
| Ensemble (background-averaged) epistasis inference | No | Yes | Yes |
| Sparse epistatic model fitting | No | No | Yes |

effects of mutations, which are the basis of molecular function. Indeed, typically a very large number of different changes in biophysical properties could underlie the same observed change in a molecular phenotype [1].

Whereas machine learning approaches and, in particular, deep learning (DL) models have had some success addressing the first limitation [2–4], they have drawbacks when it comes to interpretability and extracting mechanistic insight due to their architectural complexity. A promising alternative is to explicitly fit mechanistic models to DMS data where the model parameters are readily interpretable, corresponding, for example, to the inferred biophysical parameters (typically, but not necessarily, thermodynamic parameters) of the assayed system. This strategy has been applied to the study of gene expression regulation by transcription factors [5–11], the inference of binding affinities using ligand titrations [12–14], and to map the energetic and allosteric landscapes of protein binding domains [15, 16]. In combination with experimental designs that provide sufficient data to constrain model fitting, this can allow biophysical measurements to be made at unprecedented scale. For example, quantifying the effects of mutations on multiple molecular phenotypes and in multiple genetic backgrounds—an approach called "multidimensional mutagenesis"—has allowed us to infer changes in the folding and binding energies for tens of thousands of mutations in human protein domains [15, 16]. Neural networks provide a fast and convenient approach to quantitatively study genotype-phenotype (G-P) maps and are increasingly being used to model DMS data [7, 15–20]. Although general-use software tools for this purpose have been developed [7, 19–25], none as yet permit mechanistic models to be fit to this type of multi-phenotype and complex genotype DMS data in a flexible manner (Table 1). To address this need, we have developed MoCHI, a software tool that allows the parameterization of arbitrarily complex models using DMS data. MoCHI simplifies the task of building custom models from measurements of mutant effects on any number of phenotypes.

A further challenge is represented by the fact that the phenotypic outcome of a mutation often depends on the genetic context (or background) in which it occurs. This phenomenon, termed "epistasis" (genetic interactions), is abundant both between as well as within genes and therefore taking it into account is critical when building accurate models [26]. Epistasis comes in different flavors, either depending on the effect size of the combined mutations (global or non-specific epistasis)—manifesting as global non-linearities in the genotype–phenotype map [7, 20, 27, 28]—or their specific identities

(specific epistasis). An example of the latter is the pairwise dependency of mutation effects at physically contacting positions, for example a salt bridge between two amino acid (AA) residues that can be restored by the combination of individually disruptive mutations at the two sites [29–31]. Specific epistasis can also involve combinations of mutations at more than two positions, yet knowledge of the prevalence and origins of this "higher-order" epistasis remains limited [26]. MoCHI allows the simultaneous inference of pairwise and higher-order interaction terms (energetic couplings) for specified biophysical models facilitating deeper investigation of these phenomena. Furthermore, when a suitable user-specified mechanistic model defining the source of the global nonlinearity (epistasis) is not available, its shape can be estimated directly from the data.

Finally, quantitative definitions of epistasis vary depending on the formal concept of a genetic background (or reference) against which mutation effects (and their interactions) are calculated. The background-relative (or biochemical) view of epistasis implicitly assumes there exists a single uniquely relevant genetic background for the system under study. However, an alternative definition termed "background-averaged" epistasis—also known as "ensemble" or "statistical" epistasis—averages the effects of mutations across many different genetic backgrounds (contexts) [32]. It has been argued that this distinction is particularly important for inference within large combinatorial landscapes. In these settings, MoCHI can optionally apply mathematical theory of ensemble epistasis to learn sparse models that are both highly predictive and informative of the genetic architecture of the underlying biological system [33]. We first describe the MoCHI framework and software package for fitting interpretable genotype–phenotype models. We then demonstrate key aspects of the tool's functionality by using it to analyze a range of different DMS datasets.

## Results

### A flexible tool to fit interpretable genotype–phenotype models

MoCHI uses the data generated by DMS experiments to learn simple models that accept a genotype sequence $x$ (DNA, RNA, protein) as input and output a quantitative phenotypic prediction $\hat{y} = F(x)$ (enrichment ratio, growth rate, cellular fluorescence etc.; Fig. 1b). In contrast to DL models, the inferred parameters of the model $F$ are directly interpretable.

In the simplest scenario, the effects of single nucleotide/AA substitutions in the wild-type (reference) sequence contribute independently to the final prediction. In other words, when single mutant effects are additive in multi-mutants, the phenotypic score of an arbitrary sequence variant $i$ (of any mutational order) relative to the wild-type sequence is simply the sum over nucleotide- or residue-specific effects. This assumption is implicit in a linear model where variant sequences are one-hot encoded and the phenotypic effects of single substitutions correspond to the coefficients of the model.

However, there are two main reasons why such naïve models tend to perform poorly at the task of accurately predicting quantitative scores from DMS experiments. First, results from empirical work suggest that pairwise (and possibly higher-order) genetic interactions are abundant in biological sequences [34–36]. The mechanistic origins of specific epistasis—where the effect of a given mutation depends on the specific genetic

background in which it occurs—is an active field of research, but there is evidence that it is enriched between sequence positions involved in physical interactions [29–31, 37].

Second, the relationship between the biophysical effect of a mutation and the measured (observed) phenotype is typically nonlinear. Sources include the thermodynamics of protein folding and binding as well as the cooperativity and competition of molecular interactions such as transcription factors with DNA. Molecular phenotypes often have finite upper and lower limits, for example the quantitative level of inclusion of an alternatively spliced exon in an mRNA transcript can only occur in the range 0–100% [38]. In addition, many experimental phenotypes such as cellular growth, fluorescence intensity, gene expression level, and metabolic flux are subject to analogous upper and lower limits [26]. The consequence of these nonlinearities is that the phenotypic outcome of the sum of mutation effects at the biophysical level is not equal to the sum of their individual phenotypic effects. This "surprise" outcome of mutations—based on their magnitudes and the parameters affected but not their specific identities—is known as global epistasis or non-specific epistasis. Technical nonlinearities in the relationship between a molecular phenotype and observations thereof can also be introduced by the experimental measurement process itself [39]. Regardless of its source, it is important to properly account for global epistasis as it can dramatically improve model performance, generalizability, and interpretability, reducing the number and type of genetic interaction terms (model features/variables) required to explain phenotypic effects, a phenomenon referred to as phantom epistasis [38].

MoCHI allows the explicit simultaneous modeling of these two types of epistasis (global and specific) by formulating the genotype–phenotype map $F(x)$ as a graph consisting of sequential transformations. In the additive trait map $f(x)$, the effects of individual mutations and mutation combinations (genetic interactions) combine additively at the energetic level. The resulting sum or inferred biophysical trait $\phi = f(x)$ can be interpreted in some models as the total Gibbs free energy of the system [5–7, 12, 15, 16]. However, as this quantity is typically unobserved, it is also commonly referred to as the latent phenotype or fitness potential (Fig. 1b, left). Importantly, $f$ maps sequences to an unbounded physical quantity $\phi$ representing the ultimate mechanistic basis of mutation effects, and therefore its parameters can provide deep insight into the system under study.

The molecular phenotype $p = g(\phi) = g(f(x))$ is modeled as a nonlinear transformation of the biophysical trait $\phi$ and represents how changes at the energetic level affect the probability of molecular events, for example the fraction of molecules in a given state (folded, bound, cleaved, spliced, etc.). The mathematical formulation of the function $g$ is either determined a priori or it can be estimated directly from the data, an approach termed global epistasis (GE) regression originally developed in the evolution literature [27, 28, 39, 40] and increasingly implemented using neural networks [7, 17, 18, 20].

Finally, the observed phenotype $y = h(p) = h(g(f(x)))$ is modeled as an affine transformation (two parameter scale and shift) of the molecular phenotype $p$. The key assumption here is that the DMS experimental assay provides a quantitative score for each variant that is linearly correlated with the molecular phenotype of interest albeit on a different scale or arbitrary units. Therefore, $h$ simply represents a conversion between these two unit systems. In order to account for the uncertainty in experimental

measurements, MoCHI applies an empirical noise model by weighting the objective function with experimental error estimates when available.

A key advantage of MoCHI is the graph-like nature in which custom genotype–phenotype models are implemented as neural networks permitting an arbitrary number of measured phenotypes to report on an arbitrary number of inferred additive traits (Fig. 1b, right). Practically, this allows models to be fit simultaneously to multiple DMS datasets that result from assaying/phenotyping the same (or overlapping) variant libraries. For example, the architecture in the right panel of Fig. 1b describes two related molecular phenotypes $p_1$ and $p_2$, where each is assumed to be a different nonlinear function of two underlying biophysical traits ($\phi_1$ and $\phi_2$) as determined by the global epistasis functions $g_1(\phi_1, \phi_2)$ and $g_2(\phi_2)$. Furthermore, the molecular phenotype $p_2$ is assayed independently in two separate experiments, as denoted by $y_2$ and $y_3$, where any systematic (linear) differences between the resulting scores are captured in the inferred parameters of the transformations $h_2$ and $h_3$, thereby obviating the need for any explicit inter-experiment normalization prior to modeling.

MoCHI is implemented as a python package and relies on the PyTorch machine learning framework for model inference (Fig. 1c), with a "no-coding" option provided via a command-line tool. The package is divided into modules for DMS data management (sequence feature extraction, definition of cross-validation groups), learning tasks (model architecture definition, hyperparameter tuning and fitting), and reporting of results (model performance and diagnostics). A fourth module handles project workflows involving multiple learning tasks including sparse model inference. In what follows, we provide examples of MoCHI's functionality on various empirical datasets.

### Fitting biophysical models to DMS data with MoCHI

Although DMS involves perturbations of biomolecules at the biophysical level (by varying their sequences), functional assays of mutant effects typically involve the quantification of a convenient proxy (e.g., fluorescence score) or higher-level phenotype (e.g., cell growth rate, Fig. 1b). Mechanistic models that explicitly take into account the global nonlinear relationship between the measured (observed) phenotype and the biophysical basis of mutation effects have advantages over both linear models in terms of generalizability and DL models in terms of interpretability.

Although MoCHI imposes no restrictions on the choice/definition of the global epistasis function, equilibrium thermodynamic models provide a useful approximation of protein states under natural conditions. At thermal equilibrium, the Boltzmann distribution relates the probability that a system will be in a given state $k$ to the (Gibbs) free energy $G_k$ of the state and the temperature of the system $T$:

$$p_k = \frac{1}{Z} e^{-\frac{G_k}{RT}}$$

where $Z = \sum_{m=1}^{M} e^{-G_k/RT}$ is the partition function—with the summation over all possible states $M$ (e.g., distinct protein conformations and/or interactions)—and R is the gas constant. We first consider the simplest possible thermodynamic model of protein binding, i.e., a two-state unbound/bound model, where we denote the sum of energies of

all possible unbound states with the reference value of zero (i.e., $G_u = 0$), the fraction of molecules in the bound state is then:

$$p_b = \frac{e^{-\frac{\Delta G_b}{RT}}}{Z} = \frac{1}{1 + e^{\frac{\Delta G_b}{RT}}}$$

where $Z = 1 + e^{-\frac{\Delta G_b}{RT}}$ and $\Delta G_b$ is the energy difference between unbound and bound states. Importantly, $\Delta G_b$ implicitly depends upon the AA sequence. Figure 2a summarizes the experimental details of a DMS study in which mutations were introduced in FOS and JUN amino acid sequences, two disordered proto-oncoproteins that interact through their leucine zipper domains forming the AP-1 transcription factor complex upon heterodimerization [41]. In leucine zippers, like FOS and JUN, dimerization occurs through the formation of an alpha-helical coiled coil in a two-state process, i.e., the proteins fold upon binding and there exists no significant population of structured monomers [42, 43] (Fig. 2a). The effects of mutation combinations on protein binding were quantified by "BindingPCA" (bPCA), a method in which the proximity of two fragments of a reporter enzyme (DHFR) that are fused to the two respective proteins under study is coupled to yeast cell growth.

Figure 2c indicates a particular instance of the general genotype–phenotype framework (Fig. 1b), where a neural network is used to fit a two-state thermodynamic model (Fig. 2c, global epistasis function, $g$) to the bPCA data (Fig. 2c, bottom, target variable, $y$), thereby inferring the causal changes in free energy of binding (Fig. 2b, left inset, weight coefficients) associated with single AA substitutions (Fig. 2c, top, input values, $x$). Importantly, we assume that these energies are additive, meaning that the total free energy change ($\Delta\Delta G$) of an arbitrary variant $i$ (of any mutational order, e.g., double mutant) relative to the wild-type sequence is simply the sum over residue-specific energies ($\Delta\Delta g$) corresponding to all constituent individual (i.e., lowest order) AA changes:

$$\Delta\Delta G_{b,i} = \sum_{j=1}^{n} \Delta\Delta g_{b,j}$$

where $\Delta\Delta g_{b,j}$ denotes the binding free energy change of constituent single AA substitution $j$ of variant $i$ relative to the wild type. We can therefore express the absolute (rather than relative) free energy of binding of an arbitrary variant $i$ as:

$$\Delta G_{b,i} = \Delta g_{b,0} + \sum_{j=1}^{n} \Delta\Delta g_{b,j}$$

where $\Delta G_{b,0}$ is the binding free energy of the wild type.

The only configuration information strictly required to run MoCHI is a plain text model design file that defines the neural network architecture, and which additionally includes a path to the pre-processed DMS data for each observed phenotype (table rows), including fitness and empirical error estimates as provided by tools such as Enrich2 [45], DiMSum [46], mutscan [47], or Rosace [48] (see the "Methods" section). MoCHI conveniently handles all low-level data manipulation tasks required for model fitting including the definition of training-test-validation data splits and 1-hot encoding of sequence features from AA sequences. By default, MoCHI optimizes the trainable parameters of the neural network using the PyTorch machine learning

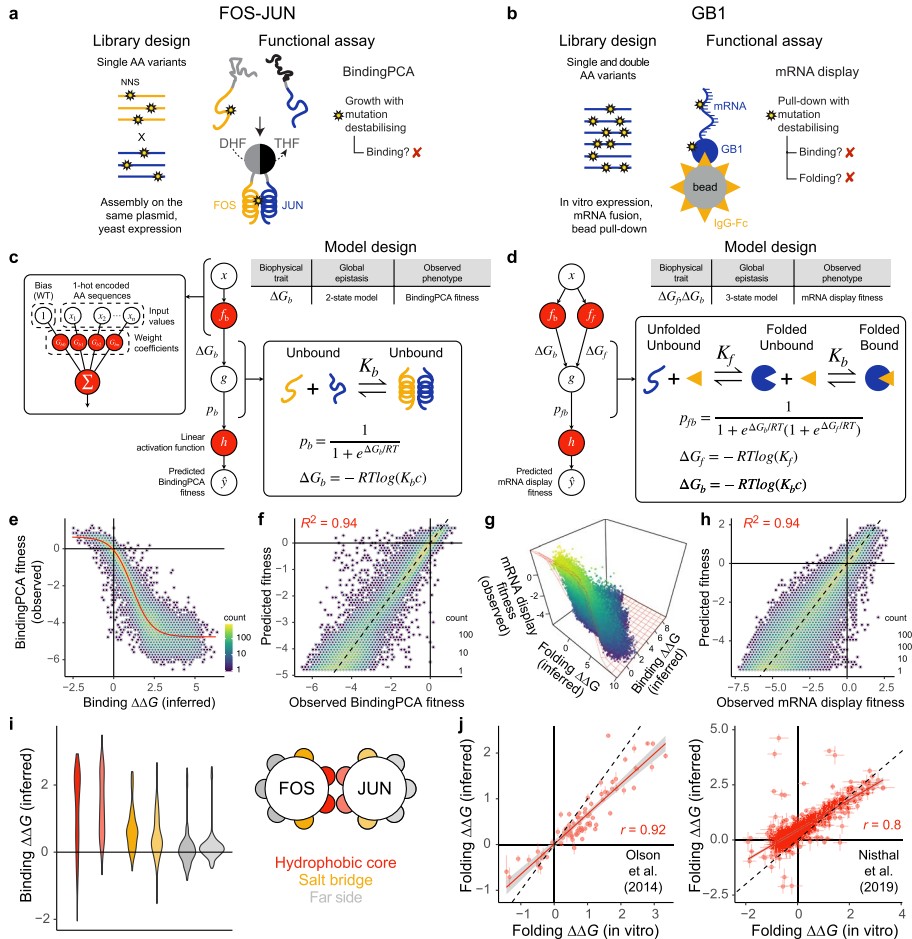

**Fig. 2** Fitting biophysical models to DMS data with MoCHI. **a** Library design and yeast growth-based functional assay used to interrogate the effects of single AA substitutions on the heterodimerization of FOS and JUN via BindingPCA (bPCA) [41]. Red cross, yeast growth defect; DHF, dihydrofolate; THF, tetrahydrofolate. **b** Library design and mRNA display-based in vitro assay used to interrogate the effects of all single and double AA substitutions in the IgG-binding domain of protein G (GB1) [31]. **c, d** Two- and three-state equilibria, thermodynamic models, neural network architectures, and corresponding MoCHI model design tables used to infer the binding and folding free energy changes ($\Delta\Delta G_f$, $\Delta\Delta G_b$) of the mutant libraries depicted in panels **a** and **b**, respectively. $\Delta G_b$, Gibbs free energy of binding; $\Delta G_f$, Gibbs free energy of folding; $K_b$, binding equilibrium constant; $K_f$, folding equilibrium constant; $c$, standard reference concentration; $p_b$, fraction bound; g, nonlinear function of $\Delta G_b$ (panel **c**) or $\Delta G_f$ and $\Delta G_b$ (panel **d**); R, gas constant; $T$, temperature in Kelvin. **e** Nonlinear relationship (global epistasis) between observed BindingPCA fitness and inferred changes in free energy of binding. Thermodynamic model fit shown in red. **f** Performance of two-state biophysical model. $R^2$ is the proportion of variance explained. **g** Nonlinear relationship between observed mRNA display fitness and inferred changes in free energies of binding and folding. **h** Performance of three-state biophysical model. **i** Violin plots showing the distributions of binding free energy changes for mutations in different structural/heptad positions in the FOS-JUN heterodimer (see legend). **j** Comparisons of confident model-inferred free energy changes to previously reported in vitro measurements [31, 44]. Error bars indicate 95% confidence intervals from a Monte Carlo simulation approach ($n = 10$ experiments). Pearson's $r$ is shown

framework and stochastic gradient descent on a loss function based on a weighted and regularized form of mean absolute error (see the "Methods" section). Optionally, the user can restrict model fitting to randomly downsampled subsets of the data and/ or variants of a given mutation order, an example of which is presented in Extended

Data Fig. 3a of Ref. [15]. Model coefficients can also be randomly downsampled. MoCHI estimates the confidence intervals of model-inferred coefficients and free energies using a Monte Carlo simulation approach (see the "Methods" section).

This extremely simple thermodynamic model provides an excellent fit to the FOS-JUN data, faithfully capturing the nonlinear relationship (global epistasis) between changes in binding free energy and the observed phenotype scores (Fig. 2e) and explaining nearly all of the variance in bPCA fitness (Fig. 2f, $R^2 = 0.94$ on held out test data), strongly supporting the assumption that most changes in free energy of binding are additive in double amino acid substitutions. Plotting the distributions of binding free energy changes separately for residues in the core, solvent-exposed surface (far side) and salt bridge positions shows that mutations in core positions comprising the hydrophobic binding interface between FOS and JUN subunits tend to have strongly destabilizing effects as expected (Fig. 2i). Salt bridge positions are also more biased towards disrupting binding than far side mutants, consistent with their structural role in stabilizing the heterodimer (Fig. 2i).

These results show that MoCHI can fit simple mechanistic models to DMS data, but to what extent do they reflect biophysical reality? How accurate are the MoCHI-inferred free energy changes? We address this question using previously published in vitro DMS data for the binding of nearly all single and double amino acid substitutions of protein G domain B1 (GB1) to IgG-Fc [31, 44] (Fig. 2b). For globular proteins like GB1, protein binding can be most simply modeled as a three-state equilibrium with unfolded, folded, and bound energetic states, where mutations can alter the concentration of the bound complex via their effects on fold stability, binding affinity, or both (Fig. 2d). The probability of the unfolded and bound state is assumed to be negligible. Although many different combinations of folding and binding free energy changes could in theory result in the same observed binding phenotype of a

(See figure on next page.)

**Fig. 3** Fitting biophysical models to DMS data assaying mutant effects on multiple phenotypes. **a** Library design and doubledeepPCA (ddPCA) functional assays (BindingPCA, bPCA and AbundancePCA, aPCA) used to interrogate the effects of all single and a subset of double AA substitutions on the cellular abundance and binding of PSD95-PDZ3 to its cognate ligand (CRIPT) [15, 16]. Green tick mark, yeast growth; red cross, yeast growth defect; DHF, dihydrofolate; THF, tetrahydrofolate. **b** Identical to panel **a** except ddPCA was applied to the oncoprotein KRAS to interrogate the effect of mutations on interactions with six different binding partners [15]. **c**, **d** Three-state equilibria, thermodynamic models, neural network architectures, and corresponding MoCHI model design tables used to infer the binding and folding free energy changes ($\Delta\Delta G_f$, $\Delta\Delta G_b$) of the mutant libraries depicted in panels **a** and **b**, respectively. Target variable predictions for the three library blocks assaying KRAS-RAF1 bPCA are depicted; the additional 15 (5 × 3) observed phenotypes corresponding to the other 5 binding partners are not shown for simplicity. $\Delta G_b$, Gibbs free energy of binding; $\Delta G_f$, Gibbs free energy of folding; $K_b$, binding equilibrium constant; $K_f$, folding equilibrium constant; $c$, standard reference concentration; $p_b$, fraction bound; $p_f$, fraction folded; $g_f$, nonlinear function of $\Delta G_f$; $g_{fb}$, nonlinear function of $\Delta G_f$ and $\Delta G_b$; R, gas constant; $T$, temperature in Kelvin. **e** Nonlinear relationship between observed bPCA fitness and inferred changes in free energies of binding and folding. Thermodynamic model fit shown in red. **f** Performance of three-state biophysical model predictions of bPCA fitness. $R^2$ is the proportion of variance explained. **g**, **h** Similar to panels **e** and **f** but corresponding to KRAS-RAF1 bPCA fitness for block 1. **i** Nonlinear relationship between observed aPCA fitness and inferred changes in free energy of binding. **j** Performance of two-state biophysical model predictions of aPCA fitness. **k**, **l** Similar to panels **e** and **f** but corresponding to KRAS aPCA fitness for block 1. **m**, **n** Comparisons of confident model-inferred free energy changes to previously reported in vitro measurements [49–51]. Error bars indicate 95% confidence intervals from a Monte Carlo simulation approach ($n = 10$ experiments). Pearson's $r$ is shown

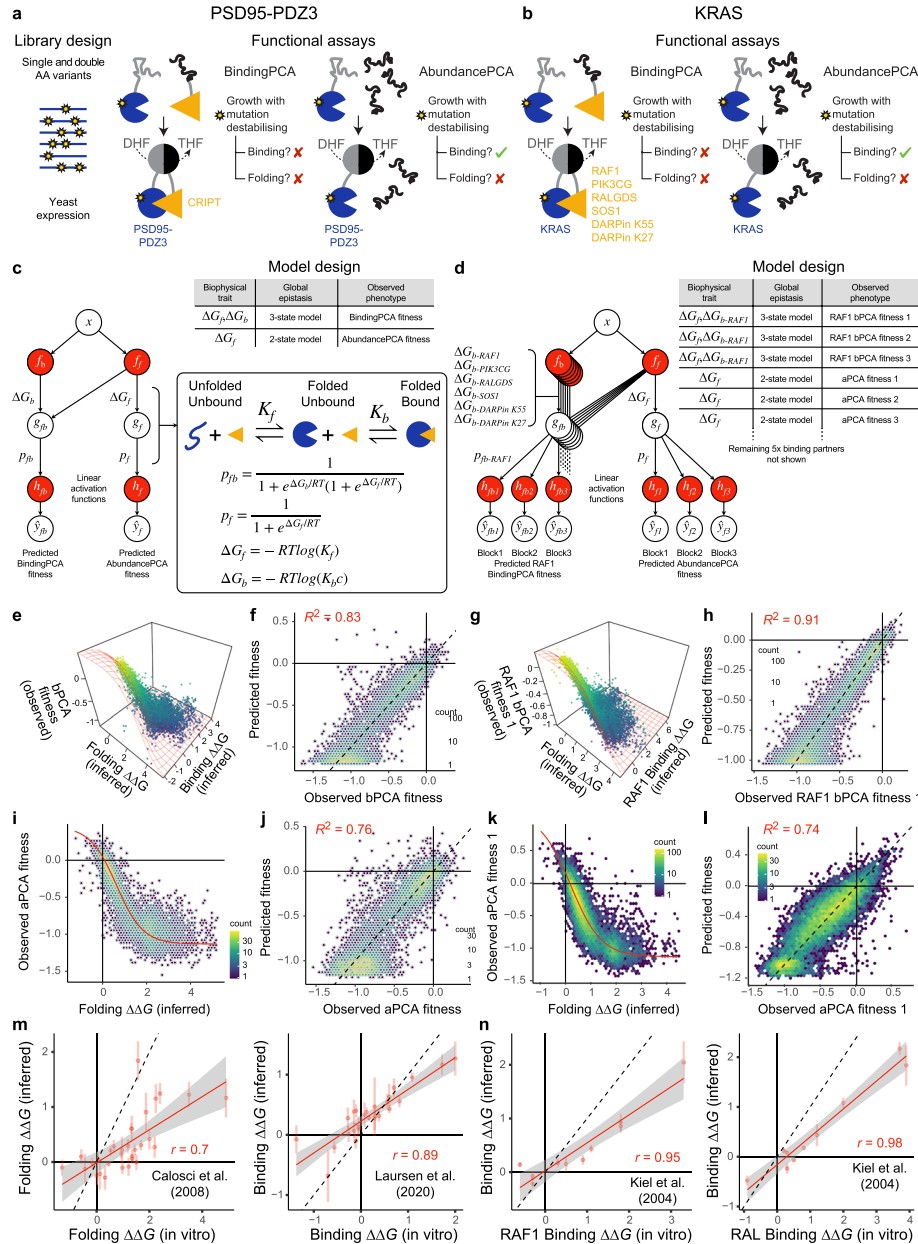

**Fig. 3** (See legend on previous page.)

particular variant, the high depth of double mutant data (singles measured in different genetic backgrounds) allows biophysical ambiguities to be resolved [1, 52].

The three-state thermodynamic model fit by MoCHI accurately predicts the binding fitness of double mutants held out during training ($R^2 = 0.94$, Fig. 2g, h), again suggesting that mutation effects overwhelmingly combine additively at the energetic (biophysical trait) level. We also find excellent agreement between the MoCHI-inferred folding free energy changes and in vitro measurements [31, 44] (Pearson's $r = 0.8$–0.92, Fig. 2j), similar to previous analyses [7, 52].

## Multimodal DMS data

The architectural flexibility of neural networks is a major advantage when dealing with more complicated DMS experimental designs. MoCHI can easily be configured to fit models to multiple DMS datasets reporting on the same or related phenotypes in which different or partially overlapping variant libraries are assayed.

We have shown previously that an approach called "multidimensional mutagenesis"—whereby the effects of mutations are quantified for multiple molecular phenotypes and in multiple genetic backgrounds—is an efficient experimental strategy to infer en masse the causal biophysical effects of mutations [15, 16]. In a similar way that double mutants are useful to constrain mechanistic models [52] (Fig. 2), measuring the effects of mutations on multiple phenotypes helps to disentangle the underlying free energy changes [1].

In Fig. 3, we summarize the results of applying a specific implementation of this approach (doubledeepPCA or ddPCA) to map the energetic and allosteric landscapes of two well-studied proteins. The effects of all singles and a "shallow" subset of double AA substitutions were quantified on binding and intracellular concentration of the free proteins using bPCA and a second related assay (AbundancePCA, aPCA) in which only one of the interacting proteins is expressed, with the other DHFR fragment being highly expressed (Fig. 3a,b). In aPCA, functional DHFR is reconstituted by random encounters, and yeast growth is proportional to the intracellular concentration of the fusion protein over more than three orders of magnitude [53]. Whereas mutations destabilizing either binding or folding result in a growth defect in bPCA, only those affecting fold stability are detrimental to cell growth in aPCA (Fig. 3a, b).

The experimental design, thermodynamic model, and neural network architecture corresponding to ddPCA applied to the third PDZ domain from the adaptor protein PSD95 (also known as DLG4) binding to the C-terminus of the protein CRIPT is depicted in Fig. 3a, c [15]. In MoCHI, additional measured phenotypes are configured by simply adding extra rows in the model design file, in this case specifying a neural network architecture where mutation effects on folding free energy are captured in a single shared biophysical trait ($\Delta G_f$) that underlies predictions of both aPCA and bPCA fitness scores (Fig. 3c). ddPCA was subsequently applied to assay the effects of > 26,000 mutations in the oncoprotein KRAS on abundance and binding to six different interaction partners: three KRAS effector proteins RAF1, PIK3CG, and RALGDS, the guanine nucleotide exchange factor (GEF) SOS1, and two DARPins, K27 and K55 [16]. KRAS mutagenesis libraries were constructed in three consecutive partially overlapping blocks along the full protein sequence and each assayed independently by aPCA and bPCA resulting in a total of $7 \times 3 = 21$ DMS datasets. Figure 3d showcases the capabilities of MoCHI in allowing models to be fit simultaneously to large numbers of DMS datasets, taking advantage of many independent measurements to constrain the inferred folding and binding free energy changes.

Although DMS data from multiple experiments corresponding to the same measured phenotype can optionally be normalized to each other explicitly before modeling, MoCHI obviates the need for this by inferring the parameters of an affine transformation between molecular phenotype and measured phenotype (Fig. 3d).

For both proteins, MoCHI fits the DMS data very well and exhibits high predictive performance on held out test data for bPCA ($R^2 = 0.83$ and 0.91 for PSD95-PDZ3 binding to CRIPT and KRAS binding to RAF1 respectively, Fig. 3e–h) and aPCA ($R^2 = 0.76$ and 0.74 for PSD95-PDZ3 and KRAS respectively, Fig. 3i–l). The high correlations of inferred binding and folding coefficients with independent in vitro measurements for PSD95-PDZ3 [49, 50] (Pearson's $r = 0.7$–$0.89$, Fig. 3m) and KRAS [51] (Pearson's $r = 0.95$–$0.98$, Fig. 3n) validates MoCHI as a tool to quantify energetic terms from DMS data.

Furthermore, by analyzing the resulting energetic landscapes in the context of structural information, specific mutations and residues enriched for large effects on binding affinity despite their distance to the binding interface can be readily identified. In both PSD95-PDZ3 and KRAS, these maps reveal both known and novel allosteric sites [15, 16]—representing the transmission of information spatially from one site to another. In addition to their biotechnological and medical value (for the prioritization of pockets for drug development), comprehensive energetic and allosteric maps provide deep insight into protein regulatory mechanisms.

### Inferring the shape of global epistasis and pairwise genetic interactions

Although mechanistic models have advantages in terms of interpretability (as shown above), the causes of global epistasis are often not well understood. In these situations, MoCHI allows global nonlinearities to be estimated directly from DMS data. We use two previously published combinatorial DMS datasets to demonstrate that global and specific sources of epistasis can be correctly distinguished and inferred simultaneously without prior hypotheses. In the first study, the authors assayed the effects on splicing of all combinations of 12 mutations that occurred during the evolution of an alternatively spliced human exon, exon 6 of the FAS gene [38]. A minigene library of $2^{10} \times 3 = 3{,}072$ exon mutant sequences was transfected and expressed in HEK293 cells, whereafter FAS exon 6 inclusion levels were determined by RT-PCR and deep sequencing (Fig. 4a) [38].

To estimate global nonlinearities, MoCHI uses a sub-network composed of a sum of sigmoidal functions (see the "Methods" section), a bottleneck architecture previously used by others [7, 17, 18]. Each consecutive layer of the sub-network—by default a single layer consisting of 20 neurons—performs a linear transformation of the outputs of the previous layer and then applies a sigmoid function to the result. Although this sub-network consists of additional trained model weights, they simply specify the parameters of a unidimensional nonlinear function $g$ mapping mutation-induced changes in the inferred additive trait (fitness potential, $\phi$) to changes in the measured phenotype (exon inclusion level, Fig. 4a). Unlike the additive trait map $f$ (Fig. 1b) whose output $\phi$ depends on the specific identities of mutations (and combinations thereof), the output of the global epistasis function $g$ depends only on the magnitude of $\phi$. Therefore, the shape of $g$ is a reflection of the peculiarities of the measured phenotype and/or functional assay rather than the underlying genetics of the system (captured by $f$).

The MoCHI-inferred nonlinear trend has a clear upper bound indicating that most assayed variants promote near-maximal exon inclusion and therefore further improvements in splicing efficiency have no (or very little) impact on measured inclusion fitness levels (Fig. 4c). This non-mechanistic model, which incorporates all 1st and 2nd order epistatic coefficients (pairwise genetic interactions), performs

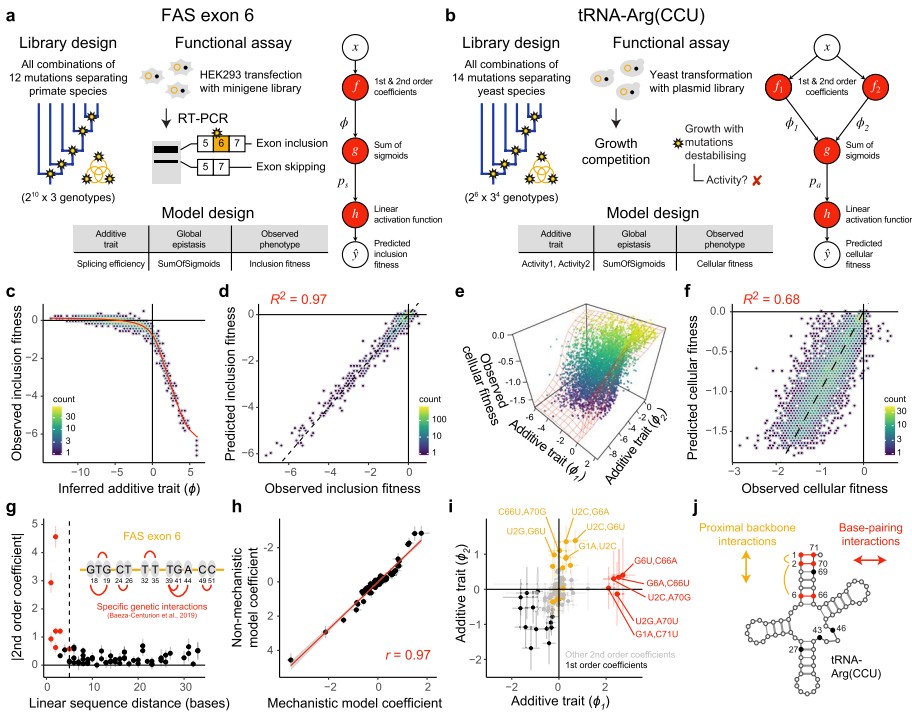

**Fig. 4** Simultaneous modeling of global and specific epistasis. **a** Library design of all combinations of 12 mutations separating FAS exon 6 in primate species, functional assay reporting on exon inclusion in HEK293 cells by mRNA sequencing [38] and neural network architecture and corresponding MoCHI model design tables used to infer the shape of global epistasis due to splice-site competition. Both 1st and 2nd order epistatic coefficients as well as a sub-network ($g(\phi)$) composed of a sum of sigmoids (see the "Methods" section) to infer unidimensional global epistasis are shown. **b** Library design of all combinations of 14 mutations separating the tRNA arginine-CCU tRNA (tRNA-Arg(CCU)) in post-whole-genome duplication yeast species, *S. cerevisiae* growth competition assay reporting on cellular fitness [34] and neural network architecture used to analyze this DMS dataset (bidimensional global epistasis). Red cross, yeast growth defect. **c** Inferred nonlinear relationship between observed inclusion fitness and the underlying additive trait ($\phi$). **d** Performance of 2nd order MoCHI model predictions of inclusion fitness. **e** Nonlinear relationship between observed cellular fitness and inferred changes in additive traits $\phi_1$ and $\phi_2$ (see panel **b**). Sum of sigmoids model fit shown in red. **f** Performance of 2nd order MoCHI model predictions of cellular fitness as a function of mutations in the tRNA-Arg(CCU). **g** The magnitude of pairwise genetic interaction terms (2nd order coefficients) versus linear sequence distance separating the individual mutated positions in FAS exon 6. Red points indicate the top specific pairwise genetic interactions as described in [38] (see inset). **h** Correlation between all 1st and 2nd order coefficients from a non-mechanistic model where global epistasis was inferred directly from the data (panel **a**) to those from a mechanistic model of splicing competition [38] (see the "Methods" section). **i** Comparison of model coefficients (1st and 2nd order, see legend and panel **j**) between inferred additive traits $\phi_1$ and $\phi_2$. Genetic interaction terms (2nd order coefficients) potentially restoring Watson–Crick base-pair interactions (see panel **j**) are indicated in red. Interactions between mutations in proximal positions (within 5 bp) that compensate individual changes in G/C content are indicated in yellow. The top 5 terms for each additive trait are labeled. **j** Secondary structure of *S. cerevisiae* tRNA-Arg(CCU) indicating variable positions (closed circles) combinatorially mutated in the DMS experiment described in panel **b**. Three Watson–Crick base pairing (WCBP) interactions involving pairs of these positions ([1, 71], [2,70] and [6,66]) are indicated in red. Two proximal G/C compensating interactions are indicated in yellow

extremely well on held-out test data ($R^2 = 0.97$, Fig. 4d), and the lack of bias in the residuals suggest that global nonlinearities in the data have been adequately accounted for by the inferred trend (Additional file 1: Fig. S1a-c). The authors of the original study used a mechanistic model of splice site competition to model the FAS exon 6 DMS data, which revealed 7 significant pairwise interactions between

mutations in neighboring positions [38]. The proximity of interacting mutations is likely due to their co-occurrence within splicing factor motifs.

Does the non-mechanistic MoCHI model recover these same findings? Comparing the magnitude of 2nd order coefficients to linear distance in the exon sequence for the model in Fig. 4a recapitulates these results where proximal mutations tend to be strongly coupled (Fig. 4g). The top 5 pairwise interactions all involve mutations within 5 bp in the linear sequence and the 7 previously reported significant pairwise interactions all occur within the top 11 in the non-mechanistic model (Fig. 4g, see inset). Lastly, we used MoCHI to fit a mechanistic model of splicing competition to the same data (see the "Methods" section), and direct comparison of the inferred coefficients between the two models shows that they are highly correlated (Pearson's $r = 0.98$, Fig. 4h, Additional file 1: Fig. S1e-h). These results demonstrate that MoCHI can simultaneously account for both global and specific epistasis even in the absence of a mechanistic hypothesis. Importantly, coefficients from a linear model do not recapitulate these results (Additional file 1: Fig. S1d,h–l).

We performed a similar analysis on a second DMS dataset where the authors assayed the effects of mutation combinations in a conditionally essential yeast gene (tRNA-Arg(CCU)) on cell growth (Fig. 4b). The library was designed to cover all 5184 ($2^6 \times 3^4$) combinations of the 14 nucleotide substitutions observed in ten positions in post-whole-genome duplication yeast species [34]. First, we specified a model with unidimensional global epistasis and up to 2nd order epistatic coefficients. This model explains roughly 50% of total fitness variance ($R^2 = 0.51$, Additional file 1: Fig. S2a).

However, with two additive traits (bidimensional global epistasis), model performance on held-out data is significantly improved (Fig. 4b, e, f, $R^2 = 0.68$). In this model, mutations and pairwise interactions have independent effects on both $\phi_1$ and $\phi_2$, and the molecular phenotype is given by $p = g(\phi_1, \phi_2)$, where $g$ is a nonlinear surface inferred from the data (Fig. 4e, see the "Methods" section). We fit models allowing for either up to third-order interactions or higher numbers of additive traits using both MoCHI and LANTERN [20], but performance was not improved in all cases (Additional file 1: Fig. S2b-f). Comparing coefficients between the two inferred additive traits ($\phi_1$ and $\phi_2$) shows that whereas first-order terms are overwhelmingly detrimental (Fig. 4i, black points), many of the top pairwise terms on $\phi_1$ are likely compensatory, involving pairs of mutations that restore Watson–Crick (or wobble) base pairing interactions that are disrupted when introduced individually (Fig. 4i, red points, Fig. 4j, red lines). On the other hand, the largest positive pairwise interactions on $\phi_2$ involve pairs of mutations that compensate for local changes in GC content (Fig. 4i, yellow points, Fig. 4j, yellow lines).

The improved performance of this model which separates interaction terms with distinct structural context into two different additive traits suggests two dominant sources of global epistasis underlying mutation effects on the measured phenotype, i.e., competent tRNA structure and function. These results are consistent with the authors' own analyses [34] although they did not attempt to control for multidimensional global nonlinearities in the data and were therefore unable to disentangle the two different mechanistic bases of pairwise mutation effects as we do here.

### Sparse models incorporating higher-order epistatic terms

A major open question in the field of synthetic biology is how important genetic interactions beyond second-order (pairwise) terms are for the task of molecular design and genetic prediction. In contrast to pairwise genetic interactions, quantifying higher-order interactions is more experimentally challenging and hence their abundance and origin is less well understood. Various methods have been developed to directly calculate epistatic coefficients from phenotypic measurements, but one particular representation, termed background-averaged (or ensemble) epistasis, has been suggested to be the most informative for constructing sparse models [32].

Recent work has shown that encoding (or embedding) sequences in this alternative basis and fitting penalized regression models allows accurate genetic prediction even in the presence of a limited number of (random) phenotypic measurements [35]. The mathematical formalism of background-averaged epistasis handling binary genetic sequences (with a maximum of two alleles per position) [54] relies on the Walsh-Hadamard transform [55, 56] but has been recently extended to handle genetic landscapes of arbitrary shape and complexity [33, 57]. MoCHI can optionally be configured to use this theory, and we demonstrate its functionally by way of reanalyzing a previously published combinatorial DMS dataset (Fig. 5a–d).

In this study, the authors used a bacterial system and FACS-seq to quantify the fluorescence brightness of all $2^{13} = 8,192$ mutants linking red and blue variants of the *Entacmaea quadricolor* fluorescent protein in order to examine epistasis up to the 13th order [35] (Fig. 5a, b). An analysis of background-averaged epistasis using both epistasis decomposition [32] and L1 (Lasso) penalized regression [35] suggests the existence of significant higher-order specific epistasis but that remarkably few interactions are required to predict the phenotypic measurements with high accuracy. We conducted a similar analysis using MoCHI by embedding amino acid sequence features in the background-averaged epistasis basis (as opposed to one-hot encoding) and including up to 6th-order epistatic terms, but without global epistasis, i.e., a penalized multiple linear regression where $g(\phi) = \phi$ (Fig. 5a). The results recapitulate the reported sparsity in epistatic coefficient space, with the model explaining almost all phenotypic variance ($R^2 = 0.93$, Additional file 1: Fig. S3a), while the vast majority of all possible 4096 terms have values near zero (Fig. 5c). Indeed, a new model including only the top 100 coefficients ranked by magnitude—including pairwise and higher-order terms up to 5th order (Fig. 5c, inset)—exhibits only a modest reduction in predictive performance ($R^2 = 0.91$, Fig. 5d).

Therefore, a relatively low number of higher-order epistatic terms are sufficient to explain the observed variance in brightness fitness, but are they required? For comparison, we fit a non-mechanistic model to the same data, allowing unidimensional global epistasis and incorporating all first and second-order epistatic coefficients. We were surprised to find that MoCHI recovers a strong global nonlinearity in the data, suggesting that brightness fitness has finite upper and lower limits dictated by either the underlying protein biophysics of fluorescence and/or due to specific technical constraints/biases intrinsic to the experimental measurement procedure (Fig. 5e). The observation that this comparatively simple model explains more phenotypic variance ($R^2 = 0.94$) with fewer epistatic terms (92 coefficients, Fig. 5f)—none of them beyond

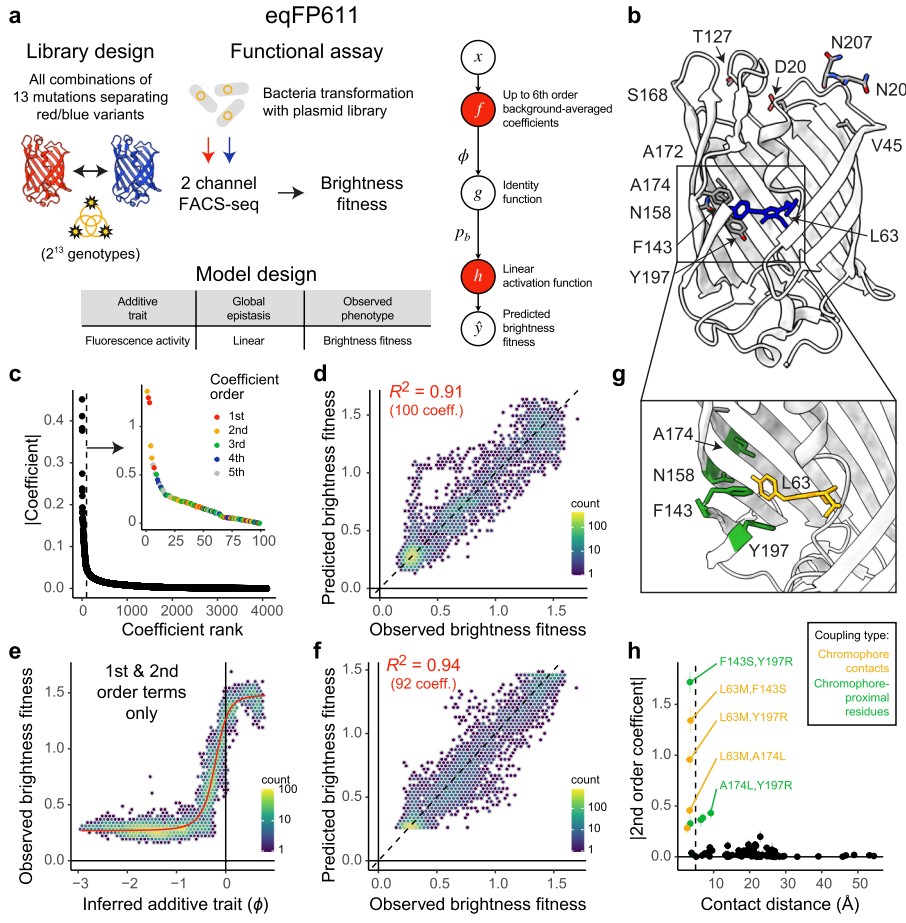

**Fig. 5** Sparse models incorporating specific higher-order epistatic terms. **a** Library design of all combinations of 13 mutations separating red and blue variants of the fluorescent *Entacmaea quadricolor* protein eqFP611, functional assay reporting on fluorescence brightness in bacteria by FACS-seq [35] and neural network architecture and corresponding MoCHI model design tables used to fit a model incorporating up to 6th order background-averaged epistatic coefficients (see the "Methods" section). **b** Crystal structure of the blue variant (TagBFP) of eqFP611 (PDB: 3M24) and 13 positions (12 shown) that differ in the red variant (mKate2) that were mutated in the DMS experiment described in [35]. **c** Epistatic coefficients (all orders) ranked by magnitude where the epistatic order of the top 100 coefficients is shown (see inset). **d** Performance of sparse MoCHI model incorporating the top 100 epistatic coefficients including terms up to 6th order. **e** Inferred nonlinear relationship between observed brightness fitness and the underlying additive trait ($\phi$) using only 1st and 2nd order epistatic terms. **f** Performance of MoCHI model incorporating global epistasis and all 92 1st and 2nd order epistatic coefficients. **g** Crystal structure of eqFP611 (PDB: 3M24) showing chromophore (yellow) and proximal mutated residues (green). Note: labeled as L63 is the N-[(5-hydroxy-1H-imidazole-2-yl) methylidene]acetamide chromophore, which is formed by the post-translational modification of the tripeptide Leu63-Tyr64-Gly65 [58]. **h** The magnitude of pairwise genetic interaction terms (2nd order coefficients) versus inter-residue distance (minimal side-chain heavy atom distance in 3D space). The top 5 coefficients are labeled. Yellow points indicate coupling terms corresponding to direct physical contacts (minimal side chain heavy atom distance < 5 Å) involving the chromophore. Green points indicate coupling terms involving pairs of chromophore-proximal residues (see panel **g**). Error bars indicate 95% confidence intervals from a Monte Carlo simulation approach ($n = 10$ experiments)

second-order—suggests that there is likely very little, if any, detectable higher-order epistasis in this dataset when properly accounting for global epistasis.

Inspecting second-order terms in this model reveals that the top 9 terms all either involve interactions between mutations of the chromophore (L63) and mutations

at physically contacting residues (minimal side chain heavy atom distance < 5 Å) or interactions between these chromophore-proximal residues (Fig. 5g, h). This illustrates the importance of taking into account global epistasis in model fitting and the utility of MoCHI when it comes to building both highly accurate and interpretable genotype–phenotype models. One caveat is that correctly inferring global epistasis requires the availability of sufficient numbers of measured variants and, practically, in situations of data sparsity, incorporating higher-order epistatic terms may result in models with improved performance (Additional file 1: Fig. S3b-e).

## Discussion

Here, we have presented MoCHI, a flexible open-source package for fitting user-specified mechanistic models to deep mutational scanning data and for simultaneously quantifying pairwise and higher-order genetic interactions between mutations (epistasis). MoCHI offers a number of important advantages over previous general-purpose software tools for modeling DMS data [7, 19, 20], as summarized in Table 1. Nn4dms [19], ECNet [22], DeepSequence [23], Tranception [24], GVP-MSA [21], and related methods [25] fit non-mechanistic (black box) models which present major challenges for interpretation. LANTERN [20] maps DMS data into low-dimensional feature space facilitating interpretation but does not fit the parameters of specified biophysical models or calculate specific interactions between variants. MAVE-NN [7] can fit biophysically interpretable models to DMS data but does not currently handle multidimensional (multimodal) phenotypes nor calculate higher-order specific interactions between variants. MoCHI addresses these limitations allowing users to fit mechanistic models to multidimensional phenotypic data. MoCHI can also learn global nonlinearities of arbitrary dimension from the data, if required.

Another key feature of MoCHI is its ability to simultaneously infer global and specific epistasis, including both pairwise and higher-order terms. We have illustrated above how this can result in much simpler—but still highly predictive—models than previous approaches and models where both the global nonlinearities and specific genetic interactions between mutations are mechanistically interpretable. For example, re-analysis of a high-dimensional fluorescent protein combinatorial mutagenesis dataset above allowed us to show that when global nonlinearities are correctly accounted for, higher-order genetic interactions are no longer required for accurate genetic prediction and protein engineering.

Based on our analyses, we expect that accurate inference of biophysical parameters from DMS data typically requires at least ten times more experimental measurements (e.g., from independent genetic backgrounds) than model coefficients. However, the impact of experimental noise, the number of phenotypes (in the case of multimodal learning), and the dimensionality of any inferred global nonlinearities on any such rule of thumb should be investigated in the future. In general, we suggest users be guided by parsimony and to favor models with fewer parameters. Identifying systematic biases in the residuals (difference between observed and fitted values) is a useful step in model evaluation. Additionally, metrics such as the Akaike information criterion (AIC) can be used to compare model quality.

## Conclusions

The combination of MoCHI and DMS allows biophysical measurements to be made at unprecedented scale. Indeed, MoCHI has allowed us to interrogate the fundamental genetic architecture of proteins, revealing it to be both simple and intelligible and allowing accurate prediction of the effects of combining many different mutations using fully interpretable thermodynamic models [36]. The combination of mutational scanning and model fitting is also allowing the systematic mapping of allosteric sites in proteins. This will allow the comprehensive identification of genetically validated surface pockets to target to inhibit or activate many different proteins important for medicine and biotechnology [15, 16].

## Methods

### Data management

MoCHI requires a table describing the neural network model design ("model_design") and including file paths to the DMS data for each measured phenotype. Figures 2, 3, 4, and 5 include examples of model design files for different modeling projects. Rows in the model design file indicate different observed phenotypes to be jointly modeled and columns specify:

- *trait*: One or more free text, comma-separated additive trait names
- *transformation*: A global epistasis function
- *phenotype*: A free text phenotype name
- *file*: A file path to variant fitness and error estimates for the corresponding phenotype

DMS data files for each phenotype include sequence variant strings ("aa_seq" or "nt_seq"), together with their observed/measured phenotype scores ("fitness") and associated empirical error estimates ("sigma"). "MochiData" objects handle data pre-processing of one or more DMS datasets for each phenotype ("FitnessData") and one-hot encoding sequence features including specified interaction terms ("max_interaction_order," Fig. 1c).

The modeling task can be restricted to a subset of variants of a given mutation order ("order_subset") and/or input variants can be downsampled to a given number or fraction of the total available if desired ("downsample_observations"). The user can also specify a subset of epistatic terms that should be fit per additive trait if required ("features"), or alternatively, coefficients can be randomly sampled without replacement up to a given number or fraction of the total available ("downsample_interactions"). An example of downsampling is presented in Extended Data Fig. 3a of Ref. [15].

MoCHI defines data splits for K-fold cross-validation according to the specified number of folds of equal size as well as the validation:test set size ratio. By default, a random 30% of variants is held out during model training, with 10% representing the test data ("k_folds = 10") and 20% representing the validation data ("validation_factor = 2"). Validation data is used to evaluate training progress and optimize hyperparameters (batch size, learning rate, $L_1$ and $L_2$ regularization penalties). Test data is used to assess final model performance. The held-out (test and validation) data can also be restricted to a

user-specified subset of variants of a given mutation order if desired ("holdout_orders"). In order to capture the uncertainty in fitness estimates, the training data is replaced with a random sample from the fitness error distribution of each variant ("training_resample = True"). The validation and test data is left unaltered. Finally, embedding sequence features in the background-averaged (ensemble) epistasis feature space is performed as previously described [33], if required ("ensemble = True").

### Neural network architecture

"MochiModel" objects make use of the PyTorch "ModuleList" class to create the neural network architecture defined in the model design table on-the-fly. The number of additive trait layers is determined by the number of unique string values in the "trait" column, the number of linear output layers is given by the number of model design table rows, and the manner in which these layers are connected is defined by the global epistasis functions specified in the "transformation" column (Fig. 1b). A key novelty of MoCHI is that it allows multiple phenotype measurements for the same (or overlapping) variant libraries to be jointly modeled. Rather than modeling a multivariate output with potential missing values, for each input sequence predictions are made for all phenotypes, and the final univariate output vector is then selected to match the source phenotype representing the target variable during optimization.

MoCHI provides a handful of global epistasis (activation) functions that can be used out-the-box ("Linear," "ReLU," "SiLU," "Sigmoid," "SumOfSigmoids," "TwoStateFractionFolded," etc.), but users are also able to supply their own custom uni- or multi-dimensional transformations in a python script following a template described in the repository documentation and supplied to MoCHI during runtime ("custom_transformations").

In order to infer global epistasis from the data, similar to previous work [7, 17, 18], by default MoCHI implements the "SumOfSigmoids" function as a sub-network consisting of one input layer, in which the number of neurons is determined by the corresponding additive trait dimensionality (1 neuron for unidimensional global epistasis, 2 for bidimensional global epistasis etc.), a single hidden layer (20 neurons), and one output layer (one neuron). All layers in the sub-network have sigmoidal activations:

$$out_i = \frac{1}{1 + e^{-input_i}}$$

The size and number of hidden layers can be customized ("sos_architecture") as well as the activation function of the output layer ("sos_outputlinear").

MoCHI assumes that nonlinearities (global epistasis) arise only in the relationship between biophysical parameters (free energies) and the molecular phenotype of interest. Nonlinearities at the level of the experimental assay can be included in the formulation of the mechanistic model (as a user-defined custom transformation) if these are known or experimentally characterized. In this way, the global epistasis model would potentially capture all sources of global epistasis. Nonlinearities at the level of the DMS assay can also potentially be inferred from the data (see Fig. 5e), but note that in this case it is not possible to identify their source, i.e., distinguish global epistasis introduced by the DMS assay from that introduced by the molecular phenotype. Inference of constituent nonlinear functions in a serial transformation represents an underdetermined problem.

### Empirical noise model

MoCHI performs model inference accounting for empirical noise ($\sigma_n$) in observed phenotype estimates ($y_n$) as supplied by the user and provided by tools such as Enrich2 [45], DiMSum [46], mutscan [47], or Rosace [48]. MoCHI can be configured to train the parameters of genotype–phenotype models assuming a Gaussian noise model:

$$P(y_n|\widehat{y}_n) = \frac{1}{\sqrt{2\pi\sigma_n^2}}exp\left(-\frac{(y_n - \widehat{y}_n)^2}{2\sigma_n^2}\right)$$

where $\widehat{y}_n$ is the predicted phenotype score of variant $n$. When empirical error estimates are not available or measurement noise is negligible, the user can supply arbitrary small "dummy" fitness errors (e.g., 1e-6), which will effectively disable empirical noise modeling.

### Loss function

Let $\theta = (\theta_f, \theta_g, \theta_h)$ denote the parameters of the genotype–phenotype model $F = h(g(f(x)))$, where $\theta_f$, $\theta_g$ and $\theta_h$ represent the parameters of the additive trait map, global epistasis function and affine transformations, respectively, as described in the first section of the Results. MoCHI optimizes the parameters $\theta$ of the neural network using stochastic gradient descent on a loss function given by:

$$L = L_{like} + L_{reg}$$

where $L_{like}$ is proportional to the negative log likelihood of the model:

$$L_{like} = -1/N\sum_{n=0}^{N-1}log[P(y_n|\widehat{y}_n)]$$

and $N$ is the batch size. $L_{reg}$ provides for regularization of $\theta_f$, the parameters of the additive trait map:

$$L_{reg} = \lambda_1||\theta_f||^2 + \lambda_2||\theta_f||^2$$

where $\lambda_1$ and $\lambda_2$ are the $L_1$ and $L_2$ regularization penalties, respectively. So, in the case of a Gaussian noise model ("loss_function_name=GaussianNLL"), the loss function is given by:

$$L[\theta] = 1/(2N)\sum_{n=0}^{N-1}(y_n - \widehat{y}_n)^2\sigma_n^{-2} + \lambda_1||\theta_f||^2 + \lambda_2||\theta_f||^2 + c$$

where $c$ is invariant in $\theta$ as it is simply a function of user-supplied empirical noise estimates:

$$c = 1/(2N)\sum_{n=0}^{N-1}log(2\pi\sigma_n^2)$$

MoCHI can alternatively use a loss function based on a weighted and regularized form of mean absolute error ("loss_function_name=WeightedL1") as described previously [15, 16]:

$$L[\theta] = 1/N \sum\nolimits_{n=0}^{N-1} |y_n - \widehat{y}_n| \sigma_n^{-1} + \lambda_1 ||\theta_f||^2 + \lambda_2 ||\theta_f||^2$$

which has a similar form to that in the case of a Gaussian noise model but is expected to be less sensitive to outliers in observed phenotype estimates and is therefore the default option. In order to penalize very large free energy changes (typically associated with extreme fitness scores), by default $\lambda_2$ is set to $10^{-6}$ ("l2_regularization_factor = 0.000001") representing light regularization.

### Model training

MoCHI performs a grid search over the supplied hyperparameter space defined by supplied lists of batch sizes ("batch_size"), learning rates ("learn_rate") and $L_1$ ("l1_regularization_factor") and $L_2$ ("l2_regularization_factor") regularization penalties. Optimal hyperparameters are defined as those resulting in the smallest validation loss after 100 training epochs ("num_epochs_grid = 100").

By default, models are then trained for a maximum of 1000 epochs ("num_epochs = 1000") using the Adam optimization algorithm with an initial learning rate of 0.05 ("learn_rate = 0.05"). MoCHI reduces the learning rate exponentially ("scheduler_gamma = 0.98") if the validation loss has not improved in the most recent ten epochs compared to the preceding ten epochs. In addition, MoCHI stops model training early ("early_stopping = True") if the WT free energy terms over the most recent ten epochs have stabilized (standard deviation $\leq 10^{-3}$).

### Uncertainties in model coefficients

For mechanistic biophysical models, free energies are calculated directly from model parameters as follows: $\Delta G = \theta RT$, where $T = 303K$ ("temperature = 30") and $R = 0.001987$ kcalK$^{-1}$ mol$^{-1}$. MoCHI estimates the confidence intervals of model-inferred coefficients and free energies using a Monte Carlo simulation approach. The variability of inferred free energy changes is calculated between separate models fit using data from (i) independent random training-validation-test splits and (ii) independent random samples of fitness estimates from their underlying error distributions (if "training_resample = True"). Users should be cautious when interpreting and comparing inferred parameters between non-mechanistic models as they may be non-identifiable unless gauge and diffeomorphic modes of models are fixed, as explained previously [7].

### Package structure

"MochiProject" objects manage an inference project/campaign, which may involve one or more inference tasks ("MochiTask") and are the entry point for the command-line tool or a typical workflow in a custom python script. "MochiTask" objects manage a collection of models ("MochiModel") for a specific training task and input dataset ("MochiData"). For example, a grid search over three batch sizes followed by tenfold cross-validation with the optimal batch size would result in a collection of 13 fitted models.

"MochiReport" objects output simple diagnostic plots to help users evaluate learning and model fit: (i) per-epoch loss curve for all cross-validation folds, (ii) observed

phenotype versus additive trait showing model fit for unidimensional epistasis, (iii) predicted versus observed model performance plot separately for each phenotype, (iv) per-epoch wild-type coefficient and residual plots for all additive traits.

### DMS datasets

FastQ files from previously published DMS experiments were re-processed with DiMSum v1.3 [46] (https://github.com/lehner-lab/DiMSum) using default settings with minor adjustments, except for the eqFP611 fluorescent protein DMS experiment for which we used the author-processed fitness estimates (brightness scores) [35]. Experimental design files and command-line options required for running DiMSum on these datasets are available on GitHub (https://github.com/lehner-lab/mochims). In all cases, adaptive minimum Input read count thresholds based on the corresponding number of nucleotide substitutions ("fitnessMinInputCountAny" option) were selected in order to minimize the fraction of reads per variant related to sequencing error-induced "variant flow" from lower order mutants.

We used MoCHI v1.1 (https://github.com/lehner-lab/MoCHI) to fit all models described here. Model design files and command-line options required for running MoCHI on these datasets are available on GitHub (https://github.com/lehner-lab/mochims). For the FOS-JUN dataset [41], we retained variants with a mean of at least 50 reads in the Input, configured unidimensional global epistasis according to a 2-state thermodynamic model, and held out a random subset of double aa mutants in the validation and test sets ("holdout_orders = 2"). The GB1 [31], PSD95-PDZ3 [15], and KRAS [16] datasets were analyzed as previously described [15, 16], i.e., fitting 3-state thermodynamic models and holding out a random subset of double aa mutants in the validation and test sets.

For the FAS exon 6 dataset [38], we fit a non-mechanistic model including all first and second-order (pairwise) interaction terms ("max_int = 2"). We fit a mechanistic model similarly, except we supplied a custom global epistasis function to match that in the original publication to model percentage spliced-in (PSI) estimates for FAS exon 6:

$$inclusion\ fitness \propto log(PSI) = log\left(\frac{k_6 A}{k_7 + k_6 A}\right)$$

where $k_6$ and $k_7$ are the splicing efficiency parameters for competing exons 6 and 7, respectively, and $A$ is the molecular effect of mutation, i.e., individual mutations introduce an $A$-fold change in splicing efficiency. To model the multiplicative (rather than additive) effect of mutation combinations, we replace $A$ with $e^\phi$ and set $k_6$ to a reference value of unity ($k_6 = 1$):

$$inclusion\ fitness \propto log(PSI) = log\left(\frac{e^\phi}{k_7 + e^\phi}\right)$$

where $\phi$ is the additive trait and $k_7$ is a global parameter inferred during model training.

For the tRNA dataset [34], we retained variants with fitness estimates in all six biological replicates (4526 variants) and fit a model including all first and second-order (pairwise) interaction terms ("max_int = 2") with bidimensional global epistasis (two

additive traits) as a sum of sigmoids with two hidden layers in the corresponding sub-network each having 20 neurons ("sos_architecture = 20,20"). For comparison, we fit models with either 3–5 inferred additive traits or including up to 3rd order interactions ("max_int = 3").

For the eqFP611 dataset [35], we first fit a Lasso regression model ("l1_regulariza-tion_factor = 0.1,0.01,0.001") including up to sixth-order ("max_int = 6") back-ground-averaged epistatic coefficients ("ensemble = True") and no global epistasis. For the sparse model, we supplied the top 100 coefficients by magnitude to MoCHI ("features") and fit a similar model without L1 regularization ("l1_regularization_factor = 0"). For comparison, we fit an alternative model including all first and sec-ond-order (pairwise) interaction terms ("max_int = 2") with unidimensional global epistasis as a sum of sigmoids with two hidden layers in the corresponding sub-net-work each having 20 neurons ("sos_architecture = 20,20"). Finally, we additionally fit these models using a randomly downsampled subset of 1000 variants from the origi-nal DMS dataset ("downsample_observations = 1000") to evaluate the effect of data sparsity on the results.

## Supplementary Information

---

Additional file 1: Supplementary figures 1-3. Supplementary figures related to Fig. 4 and Fig. 5.

Additional file 2: Table S1. Supplementary Table 1. Inferred additive trait parameters (and free energies) from all models.

Additional file 3. Review history.

---

### Acknowledgements
We thank all members of the Lehner Lab for helpful discussions and suggestions.

### Peer review information

### Review history
The review history is available as Additional file 3.

### Authors' contributions
B.L. and A.J.F. conceived the project. A.J.F. conducted software development and data analysis. A.J.F. wrote the first draft of the manuscript which was edited by B.L.

### Funding
This work was funded by European Research Council (ERC) Advanced grant (883742), the Spanish Ministry of Science and Innovation (LCF/PR/HR21/52410004, EMBL Partnership, Severo Ochoa Centre of Excellence), the Bettencourt Schuel-ler Foundation, the AXA Research Fund, Agencia de Gestio d'Ajuts Universitaris i de Recerca (AGAUR, 2017 SGR 1322), and the CERCA Program/Generalitat de Catalunya. A.J.F. was funded by a Ramón y Cajal fellowship (RYC2021-033375-I) financed by the Spanish Ministry of Science and Innovation (MCIN/AEI/https://doi.org/10.13039/501100011033) and the European Union (NextGenerationEU/PRTR).

### Data availability
Source code of the general-purpose software tool (Python package) that is the topic of this manuscript (MoCHI v1.1) is available at https://github.com/lehner-lab/MoCHI. All other source code including DiMSum and MoCHI configuration files, scripts to perform all downstream analyses of MoCHI-fitted models, and model comparisons to reproduce all figures in the manuscript is available at https://github.com/lehner-lab/mochims. An archive of this repository is also publicly available on Zenodo at https://zenodo.org/doi/https://doi.org/10.5281/zenodo.13285580 [59]. All software is released under the MIT License, which permits unrestricted use, modification, and distribution. Inferred additive trait parameters (and free energies) from all models are provided in Table S1. The FOS-JUN DMS data is available with GEO accession GSE102901 [60]; the GB1 DMS data is available in Table S2 of Olson et al. 2014 [31]; the PSD95-PDZ3 DMS data is available with GEO accession GSE184042 [61]; the KRAS DMS data is available with SRA accession PRJNA907205 [62]; the FAS exon 6 DMS data is available with GEO accession GSE111316 [63]; the tRNA-Arg(CCU) DMS data is available with GEO acces-sion GSE99418 [64]; the eqFP611 DMS data is available with SRA accession PRJNA560590 [65].

**Declarations**
Ethics approval and consent to participate
Ethical approval was not needed for the study.

**Competing interests**
A.J.F. and B.L. are founders, employees, and shareholders of ALLOX.

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

## 