## [Additional file 3. Review history. · Genome Biology]

1st round

Reviewer 1

This manuscript by Faure and Lehner introduces MoCHI, a machine learning software designed for building custom models from deep mutational scanning (DMS) data to account for different phenotypes. MoCHI enables users to infer changes in free energy, pairwise/high-order epistasis, and more.

Compared to previous methods, MoCHI has two main advantages:

1. It allows users to fit predefined mechanistic models to multidimensional phenotypic data.
2. It can infer epistasis, encompassing both pairwise and higher-order terms simultaneously.

Given the rapid accumulation of DMS data from diverse sources, there's an urgent need for accurate models that can predict the effects of combining many different mutations. These models should be better interpretable or have biophysical meaning. This study is well-designed to address these challenges. The manuscript is clearly written, making the model strategy and process easy to follow. I have the following comments that should be addressed before considering the publication of the manuscript.

We thank the referee for their enthusiasm and very constructive comments.

Major:

(1) A significant limitation of mechanistic models is the requirement to make hypothesis about the mechanism or relationship between the observed phenotype and the biophysical basis of mutation effects. While a heuristic approach can aid in identifying plausible mapping relationships, I doubt whether it will surpass "black-box" machine learning or deep learning models, particularly regarding generalization performance. I would recommend that the authors add more in-depth comparisons of existing machine learning models' performance in the revision. The manuscript already mentions some models, such as nn4dms, LANTERN, and MAVEN-NN. However, the authors should also cite and compare models like GVP-MSA (Cell Systems 14, 706-721), which can also fit multidimensional phenotypic data from various DMS studies of different proteins, just like MoCHI. Additionally, the authors should consider models including EC-Net (Nat. Commun. 12.5743), and DeepSequence (Nat. Methods 15, 816-822), etc.

Thank you for this suggestion. In the revised manuscript we have included a table summarising the capabilities of MoCHI and comparing them to previous approaches (Table 1). We have also highlighted these differences in the Discussion and added citations to additional methods, including all of those suggested by the referee.

“MoCHI offers a number of important advantages over previous general-purpose software tools for modeling DMS data[7,19,20], as summarized in Table 1. Nn4dms[19], ECNet[22], DeepSequence[23], Tranception[24], GVP-MSA[21], and related methods[25] fit non-mechanistic (black box) models which present major challenges for interpretation. LANTERN[20] maps DMS data into low-dimensional feature space facilitating interpretation, but does not fit the parameters of specified biophysical models or calculate specific interactions between variants. MAVEN-NN[7] can fit biophysically interpretable models to DMS data, but does not currently handle multidimensional (multimodal) phenotypes nor calculate higher-order specific interactions between variants. MoCHI addresses these limitations allowing users to fit mechanistic models to multidimensional phenotypic data. MoCHI can also learn global nonlinearities of arbitrary dimension from the data, if required.”

(2) The impact of making incorrect or unreasonable prior hypothesis about the protein regulatory mechanism on the model's performance is not clearly investigated, which is undoubtedly very common. For instance, if we alter our hypothesis about the mechanism in Figure 2A - that is, if we consider that the effects of mutation combinations on protein binding don't adhere to the "unbound-bound" two-state process - would this reduce the model's generalization capability? I recommend that the author incorporate more controls into the experiments depicted in Figure 2 and Figure 3. This would allow for a more thorough analysis of the impact that varying mechanistic hypotheses have on the model's generalization performance.

We have added a paragraph to the discussion section with some general recommendations about this:

“Based on our analyses, we expect that accurate inference of biophysical parameters from DMS data typically requires at least ten times more experimental measurements (e.g. from independent genetic backgrounds) than model coefficients. However, the impact of experimental noise, the number of phenotypes (in the case of multimodal learning) and the dimensionality of any inferred global nonlinearities on any such rule of thumb should be investigated in the future. In general, we suggest users be guided by parsimony and to favor models with fewer parameters. Identifying systematic biases in the residuals (difference between observed and fitted values) is a useful step in model evaluation. Additionally, metrics such as the Akaike information criterion (AIC) can be used to compare model quality.”

(3) Unfortunately, the models based on the mechanistic hypothesis did not outperform the straightforward non-mechanistic models that only consider 1st and 2nd-order epistatic coefficients, as shown in Figure 4h and Figure 5f. These findings further raise my concerns mentioned in Comment #1, which indicate that making hypothesis about unknown biological mechanisms may not necessarily improve the model's performance.

In Fig. 4h the ‘mechanistic model’ (Baeza et al., Cell 2024) and the ‘non-mechanistic model’ (fitted here) have extremely similar forms (sigmoidal global epistasis). We included this comparison precisely to illustrate the point that in the absence of a mechanistic model MoCHI can be used to infer the form of global epistasis.

The comparison between Fig. 5e-f and Fig. 5c-d is to make the point that the previous claim that this dataset provides evidence for higher order epistasis is not supported when global epistasis (the nonlinear relationship between energy and the measured phenotype) is included in a model: a model that incorporates global epistasis and pairwise energetic couplings predicts that data just as well as the more complicated published model with higher order epistatic terms.

(4) Another disappointing result is that although the model can consider higher-order epistatic terms, they do not perform better than simple models that only consider 1st and 2nd-order epistatic coefficients. Therefore, from a model architecture standpoint, incorporating mechanistic hypotheses and higher-order coefficients hasn't yielded additional benefits. In other words, it seems that the model primarily predicts phenotypic data by learning residue-residue or base-pairing interaction terms. Existing deep learning models based on Multiple Sequence Alignment (MSA) or three-dimensional structure may have more significant advantages in capturing these pairwise interactions.

With respect, we disagree with this sentiment. In Fig. 5 we illustrate that MoCHI can be used to fit higher order models and we also illustrate that in this case there is no evidence that such complicated

models are required to make good predictions. Rather our analyses suggest the previous claim that was made that this dataset presents evidence for the importance of higher order epistasis is not supported by the data: simple pairwise energetic couplings that are highly consistent with structural information are sufficient. We included this comparison precisely to illustrate how MoCHI can be used to compare more complicated and simpler models and to illustrate that potentially misleading conclusions that can be drawn when nonlinearities in genotype-phenotype datasets are not correctly accounted for.

Minor:

The residue L63, as labelled in Figure 5b and 5g, does not appear to be a leucine based on the displayed structure. Please confirm if it is correct.

Thanks for spotting this. The residue labeled as L63 in Fig. 5b,g is the N-[(5-hydroxy-1H-imidazole-2-yl)methylidene]acetamide chromophore, which is formed by the post-translational modification of the tripeptide Leu63-Tyr64-Gly65. We have added a note and reference in the legend of Fig. 5 to clarify this.

Reviewer 2

The authors present a new code package (MoCHI) for fitting deep mutational scanning (DMS) data to obtain biophysical parameters using a neural network. The approach is quite flexible and can also learn mechanism-free non-linear relationships from the data. They have used this approach in their own prior work with great success and this software tool is designed to allow others to more easily apply their method. Overall the paper is well written, clear, and we anticipate that MoCHI will find broad use among the DMS and multiplexed analysis of variant effect (MAVE) community. However, there are a few places in the manuscript where it would help to provide additional guidance on use and interpretation. Likewise, there is some ambiguity about the github repo which would be helpful to address.

We thank the referee for their enthusiasm and very constructive comments.

Major Comments:

- * Readers and potential users would benefit from more guidance on application and data interpretation. In particular:
 - o How can one assess if you have "enough" data to constrain the model? Is there any rule of thumb regarding the number of experimental measurements vs inferred parameters?

We have added the following sentence to the Discussion:

“Based on our analyses, we expect that accurate inference of biophysical parameters from DMS data typically requires at least ten times more experimental measurements (e.g. from independent genetic backgrounds) than model coefficients. However, the impact of experimental noise, the number of phenotypes (in the case of multimodal learning) and the dimensionality of any inferred global nonlinearities on any such rule of thumb should be investigated in the future.”

- o Does the package provide any (bootstrapped or similar) estimates of confidence in the inferred parameters?

Yes, this is described in the ‘Uncertainties in model coefficients’ section of the Methods:

“MoCHI estimates the confidence intervals of model-inferred coefficients and free energies using a Monte Carlo simulation approach. The variability of inferred free energy changes is calculated between separate models fit using data from (i) independent random training-validation-test splits and (ii) independent random samples of fitness estimates from their underlying error distributions (if ‘training_resample = True’).”

o Related to this, on page 13, the authors write "the high depth of double mutant data (singles measured in different genetic backgrounds) allows biophysical ambiguities to be resolved". It might help to show an example of what the data looks like/how one might diagnose biophysical ambiguities if the data are NOT high enough depth by subsampling the data and repeating the fit. Again, some guidance on how to diagnose errors, or troubleshoot poorly fit data would help.

We provided three examples of this in Faure et al. 2022, which we now reference from the Methods (“Downsampling double mutant data in both datasets illustrates how increasing the number of double mutants improves the model fit (Extended Data Fig. 3a)”). For more general discussion of ‘biophysical ambiguities’ please see Reference 1 (<https://www.nature.com/articles/s41467-020-18694-0>).

o Likewise, are there any signatures of choosing an inappropriate biophysical model? We found the analysis of the fluorescent protein data very interesting in this regard, can you comment a bit on criteria for model selection (is it really just looking at fit quality and balancing this against the number of parameters? Should one consider a statistical measure to justify adding more parameters similar to AIC analysis?)

Thank you for this suggestion. We have added the following comment to the Discussion:

“In general, we suggest users be guided by parsimony and to favor models with fewer parameters. Identifying systematic biases in the residuals (difference between observed and fitted values) is a useful step in model evaluation. Additionally, metrics such as the Akaike information criterion (AIC) can be used to compare model quality.”

* Our understanding is that the model allows for a non-linearity at the level of g (global epistasis acting on some additive trait map), but that the experimental assay (h) is always treated as a linear effect. However, we can imagine cases where you might have a non-linear effect at the assay step as well (and you may even be able to experimentally characterize/define the non-linearity). How would one capture this with MoCHI? Does/can the model allow for non-linearities to be included at both g and h?

This is a good point. Practically, nonlinearities at the level of the experimental assay could be included in the formulation of g (as a user-defined custom transformation) if these are known/characterized. In this way, g would potentially capture all sources of global epistasis. To maintain the conceptual distinction between nonlinearities from different sources, in future updates to MoCHI we will consider allowing customisable nonlinearities in both g and h, although the mathematical form would be identical to combining them.

We have added the following paragraph to the Methods section:

“MoCHI assumes that nonlinearities (global epistasis) arise only in the relationship between biophysical parameters (free energies) and the molecular phenotype of interest. Nonlinearities at the level of the

experimental assay can be included in the formulation of the mechanistic model (as a user-defined custom transformation) if these are known or experimentally characterized. In this way, the global epistasis model would potentially capture all sources of global epistasis. Nonlinearities at the level of the DMS assay can also potentially be inferred from the data (see Fig. 5e), but note that in this case it is not possible to identify their source i.e. distinguish global epistasis introduced by the DMS assay from that introduced by the molecular phenotype. Inference of constituent nonlinear functions in a serial transformation represents an underdetermined problem.”

* We tried to download and use the code, however we ran into an ambiguity about which code repository to use. There are 2 different MoCHI repositories on github, both referenced in the methods section: <https://github.com/lehner-lab/mochims> (page 30, line 47); does not include any data in "Required Data" ("here" doesn't contain anything). And also this repository - <https://github.com/lehner-lab/MoCHI> (page 30, line 45) which appears more complete. Are there distinct roles for each repo? If not, which should users follow? Clarifying this is essential to usage and reproducibility.

The second repository (<https://github.com/lehner-lab/MoCHI>) provides the MoCHI code, i.e. the general-purpose software tool (Python package) that is the topic of this manuscript.

The first repository (<https://github.com/lehner-lab/mochims>) includes the code to perform all downstream analyses of MoCHI-fitted models, model comparisons and to reproduce all figures in the manuscript. An archive of this repository is also publicly available on Zenodo at <https://zenodo.org/doi/10.5281/zenodo.13285580>. We have now clarified this difference in the “Availability of data and materials” section. We also corrected the link to the missing “Required Data” already while this manuscript was under revision, but unfortunately not in time for this referee.

* The authors briefly mention a few other methods (LANTERN, nn4dms, MAVE-NN), however the discussion section commenting on the relationship between these is very brief. Can they provide a little more description of what the other methods are for, and how they relate to MoCHI? As currently written, it is not accessible to readers who lack fairly extensive prior knowledge of these other methods.

Thank you for this suggestion. In the revised manuscript we have included a table summarizing the capabilities of MoCHI and comparing them to previous approaches (Table 1). We have also highlighted these differences in the Discussion and added citations to additional methods, including all of those suggested by the referee.

“MoCHI offers a number of important advantages over previous general-purpose software tools for modeling DMS data[7,19,20], as summarized in Table 1. Nn4dms[19], ECNet[22], DeepSequence[23], Tranception[24], GVP-MSA[21], and related methods[25] fit non-mechanistic (black box) models which present major challenges for interpretation. LANTERN[20] maps DMS data into low-dimensional feature space facilitating interpretation, but does not fit the parameters of specified biophysical models or calculate specific interactions between variants. MAVE-NN[7] can fit biophysically interpretable models to DMS data, but does not currently handle multidimensional (multimodal) phenotypes nor calculate higher-order specific interactions between variants. MoCHI addresses these limitations allowing users to fit mechanistic models to multidimensional phenotypic data. MoCHI can also learn global nonlinearities of arbitrary dimension from the data, if required.”

Minor Comments:

* Page 11, lines 22-27 the authors reference figure 2B, but we believe they mean 2C

Corrected - thank you.

* Page 16, lines 45, figures 3m and 3n are swapped when referenced

Corrected - thank you.

Reviewer 3

In this work, Faure and Lehner propose the MoCHI framework to train interpretable models that predict phenotypes from sequence on the output of Multiplexed Assays of Variant Effect (MAVEs). The MoCHI framework models the relationship between genotype and phenotype as a composition of three functions, an additive trait map that models underlying biophysical quantities as the sum of contributions from each sequence position and their interactions, a global epistasis model that models the nonlinear relationship between free energies and the molecular phenotypes of interest, and an affine transformation from the molecular phenotype to the experimental output. Faure and Lehner provide multiple example use cases covering a range of experimental designs. They first fit basic biophysical models to deep mutational scanning (DMS) datasets, and then apply MoCHI to multidimensional phenotypic readouts by fitting models to datasets which measure both the stability and binding affinity of the same protein sequences. They then fit models to measurements of complex sequence libraries to demonstrate MoCHI's ability to model global epistasis and specific interactions between variants simultaneously, and finally fit models that incorporate up to 6-way interactions to demonstrate its ability to model higher-order interactions. In all cases, the MoCHI models achieve high predictive power.

The work by Faure and Lehner introduces a promising approach for sequence-to-function modeling. The examples in the manuscript demonstrate that MoCHI can achieve high predictive power on a broad range of DMS datasets. In the examples given, the authors are able to extract convincing biophysical interpretations from the MoCHI models. MoCHI effectively builds on existing approaches for fitting models to DMS data by enabling the modeling of multiple phenotypes and higher-order interactions.

We thank the referee for their enthusiasm and very constructive comments.

Below I provide three major critiques and some minor critiques of the paper, to help the authors improve the quality of the manuscript.

Major points

1. I was not able to run all of the examples in the MoCHI github. I could not run the example model design given under "Option A" or the example script given under "Option B". While I was able to run the demo script "demo_mochi.py", I could only do so after cloning the github repository, which the installation instructions recommend against in favor of installing through conda. When I installed using conda, none of the demos ran.

We have now updated the GitHub documentation, clarified that usage Option A requires creating a valid model design file and tested that both usage Option B and the Demo run without errors with a working installation.

2. The MoCHI framework makes a major assumption that nonlinearity arises only in the relationship between biophysical parameters (free energies) and the molecular phenotype of interest, and that the experimental measurement can be modeled with just an affine transform. The authors should justify why this is an acceptable modeling choice, given that many DMS assays provide readouts that are highly nonlinear functions of the underlying molecular phenotype.

This is a fair point. Practically, nonlinearities at the level of the experimental assay could be included in the formulation of the mechanistic model (as a user-defined custom transformation) if these are known or experimentally characterized. In this way, the global epistasis model would potentially capture all sources of global epistasis. To maintain the conceptual distinction between nonlinearities from different sources, in future updates to MoCHI we will consider allowing customisable nonlinearities from different sources, although the mathematical form would be identical to combining them.

When potential nonlinearities at the level of the DMS assay are not known, the composite global epistasis function (result of serial transformations) can potentially be inferred by MoCHI (see for example Fig. 5e). Separately inferring global epistasis introduced by the DMS assay and the molecular phenotype (constituent nonlinear functions in the serial transformation) is not possible without measuring the molecular phenotype of interest as this represents an underdetermined problem.

To clarify this, we have added the following paragraph to the Methods section:

“MoCHI assumes that nonlinearities (global epistasis) arise only in the relationship between biophysical parameters (free energies) and the molecular phenotype of interest. Nonlinearities at the level of the experimental assay can be included in the formulation of the mechanistic model (as a user-defined custom transformation) if these are known or experimentally characterized. In this way, the global epistasis model would potentially capture all sources of global epistasis. Nonlinearities at the level of the DMS assay can also potentially be inferred from the data (see Fig. 5e), but note that in this case it is not possible to identify their source i.e. distinguish global epistasis introduced by the DMS assay from that introduced by the molecular phenotype. Inference of constituent nonlinear functions in a serial transformation represents an underdetermined problem.”

3. The authors do not provide any side-by-side comparisons between MoCHI and preexisting methods. Without such comparisons, it can be difficult to determine to what extent MoCHI aids in the modeling of the example datasets (beyond what previous methods have allowed)

Thank you for this suggestion. In the revised manuscript we have included a table summarizing the capabilities of MoCHI and comparing them to previous approaches (Table 1). We have also highlighted these differences in the Discussion and added citations to additional methods, including all of those suggested by the referee.

“MoCHI offers a number of important advantages over previous general-purpose software tools for modeling DMS data[7,19,20], as summarized in Table 1. Nn4dms[19], ECNet[22], DeepSequence[23], Tranception[24], GVP-MSA[21], and related methods[25] fit non-mechanistic (black box) models which present major challenges for interpretation. LANTERN[20] maps DMS data into low-dimensional feature

space facilitating interpretation, but does not fit the parameters of specified biophysical models or calculate specific interactions between variants. MAVE-NN[7] can fit biophysically interpretable models to DMS data, but does not currently handle multidimensional (multimodal) phenotypes nor calculate higher-order specific interactions between variants. MoCHI addresses these limitations allowing users to fit mechanistic models to multidimensional phenotypic data. MoCHI can also learn global nonlinearities of arbitrary dimension from the data, if required.”

Minor points

1. Line 38 of page 5: I would recommend citing additional papers that combine libraries of regulatory sequence variants and biophysical modeling. Some possibilities are:
Mogno, I., Kwasnieski, J. C. & Cohen, B. A. Massively parallel synthetic promoter assays reveal the in vivo effects of binding site variants. *Genome Res.* (2013).
Belliveau, N. M. *et al.* Systematic approach for dissecting the molecular mechanisms of transcriptional regulation in bacteria. *PNAS* (2018).

Gertz, J., Siggia, E. D. & Cohen, B. A. Analysis of combinatorial cis-regulation in synthetic and genomic promoters. *Nature* (2009) [like Forcier et al. (2016) this does not use deep sequencing, but it does biophysically model sequence-function relationships].
Fiore, C., & Cohen, B. A. Interactions between pluripotency factors specify cis-regulation in embryonic stem cells. *Genome Res* (2016).

Thank you for these suggestions - references added.

2. Line 38 of page 5: I would recommend citing additional papers that infer binding affinities through ligand titrations. Some possibilities are:
Starr, T. N. *et al.* Deep mutational scanning of SARS-CoV-2 receptor binding domain reveals constraints on folding and ACE2 binding. *Cell* (2020).
Phillips, A. M. *et al.* Binding affinity landscapes constrain the evolution of broadly neutralizing anti-influenza antibodies. *eLife* (2021).

Thank you for these suggestions - references added.

3. Line 30 of page 6: I am skeptical of the claim that background-averaged is less sensitive than reference-sequence-based epistasis. Reference-sequence-based models and background-averaged models of epistasis mathematically equivalent, as their parameters are related by a gauge transformation (a transformation of model parameters that does not affect model predictions). A relevant reference is:

Dupic, T., Phillips, A. M. & Desai, M. M. Protein sequence landscapes are not so simple: on reference-free versus reference-based inference. *bioRxiv* (2024) doi:10.1101/2024.01.29.577800.

We agree and have deleted this clause.

4. Line 39 of page 9: “MoCHI applies an empirical noise model by weighting the objective function with experimental error estimates when available.” The authors should discuss whether and how this model is usable when empirical error estimates are not available.

When empirical error estimates are not available or measurement noise is negligible, the user can supply arbitrary small 'dummy' fitness errors (e.g. 1e-6), which will effectively disable empirical noise modeling. We now mention this alternative in the Methods.

5. Line 37 of page 10: The equation $P_k \propto e^{-G_k/RT}$ technically is not correct, because the proportionality constant also depends on G_k . I suggest $P_k = \frac{1}{Z} e^{-G_k/RT}$ with $Z = \sum_m e^{-G_m/RT}$, Z being the partition function, instead.

We thank the reviewer for pointing out this error and have now incorporated the suggested correction.

6. Line 45 of page 10: It is unnecessary to use $\frac{S_k}{S}$; it can be denoted simply as P_k

We agree and have removed this unnecessary formula.

7. Line 54 of page 11: "The only configuration information strictly required to run MoCHI is a plain text model design file that defines the neural network architecture, and which additionally includes a path to the pre-processed DMS data for each observed phenotype (table rows) as provided by tools such as Enrich2, DiMSum, mutscan or Rosace (see Methods)." The authors should explain what file formats are output by these methods, and thus are used by MoCHI.

We now realize this sentence is ambiguous and apologize for the confusion: MoCHI does not accept raw output files as provided by these tools. We have now corrected the sentence (see below) and updated the GitHub README with input file examples (<https://github.com/lehner-lab/MoCHI>):

"...a path to the pre-processed DMS data for each observed phenotype (table rows), including fitness and empirical error estimates as provided by tools such as Enrich2[44], DiMSum[45], mutscan[46] or Rosace[47] (see Methods)."

8. Lines 35-45 of page 19: "Comparing coefficients between the two inferred additive traits" The authors should elaborate on how these two additive traits are being modeled.

We have now included a description of how these two additive traits are being modeled and refer the reader to the updated Methods section:

"In this model, mutations and pairwise interactions have independent effects on both ϕ_1 and ϕ_2 , and the molecular phenotype is given by $p = g(\phi_1, \phi_2)$, where g is a nonlinear surface inferred from the data (Fig. 4e, see Methods)."

9. Line 23 of page 25: "Similar to previous work" The authors might also wish to cite refs 7 and 11 here.

Thank you - references added.

2nd round

Reviewer 1

Reviewer #1: My previous comments and suggestions have been addressed. I'd like to recommend the acceptance of this manuscript.

Thank you.

Reviewer 2

Thank you for a nice revision. I especially appreciate that the organization of the github repositories has been clarified. Though it is a matter of taste, I do think it would be useful to give a couple of sentences in the main text regarding: (1) model fit in response to downsampling the data (with the reference to Faure 2022), and (2) uncertainty estimation for model coefficients. I appreciate that both of these items are addressed in the methods, but to me, they are sufficiently important to future users that they should at least be referred to in the main text.

We have now added the following to the second subsection of the Results:

“Optionally, the user can restrict model fitting to randomly downsampled subsets of the data and/or variants of a given mutation order, an example of which is presented in Extended Data Fig. 3a of Ref. [15]. Model coefficients can also be randomly downsampled. MoCHI estimates the confidence intervals of model-inferred coefficients and free energies using a Monte Carlo simulation approach (see Methods).”

Reviewer 3

The authors have satisfactorily addressed our previous comments. We have just three additional points that the authors may wish to consider before publication:

1. The example for option A in their github usage guide still doesn't work because the example variant fitness files they provide are different from the ones used in their example model design. All the other examples they provide work as presented now.

Thank you, this has now been fixed.

2. Their advice to favor models with fewer parameters applies to classical regression models, but is questionable for neural nets, which often benefit from overparameterization (i.e., the phenomenon of double descent). This is a minor point because most of the architectures they present are either biophysical models or generalized linear models with interactions.

Given the purpose of the software tool and models presented, we believe this point is beyond the scope of this work.

3. I believe that MAVE-NN does allow one report parameters that represent background-averaged epistasis via use of the "uniform" gauge. This, however, only affects the second-to-last line of Table 1.

Thank you, we have now updated Table 1 to indicate MAVE-NN has this capability.